# Mechano-bactericidal activity of cicada wing nanostructures against gram-positive bacteria

Jianwei Qu,[1] Shiya Gu,[1] Lei Chen,[1] Fangming Cui,[1] Huan Wang,[1] Liyan Wu[1]

**ABSTRACT** Traditional antifouling mechanisms primarily prevent biofouling by inhibiting the initial adhesion of microorganisms to material surfaces. Conversely, the antifouling effect of cicada wings arises from their microstructured or nanostructured surface. These structures trap and rupture adherent microorganisms, which prevent biofilm formation and enable effective antifouling. This study used two typical gram-positive pathogenic strains—the rod-shaped *Bacillus cereus* and the coccus-shaped *Staphylococcus aureus*—as model microorganisms. The bactericidal efficacy of the nanopillar array on *Pomponia linearis* cicada wings against surface-adhered gram-positive bacteria was quantitatively evaluated using live/dead staining. Additionally, the interfacial morphological evolution during bacteria–structure interaction was visualized via scanning electron microscopy/transmission electron microscopy, aiming to elucidate how physical disruption compromises bacterial cellular integrity. The results show that the nanopillar surface exhibits potent bactericidal activity against both gram-positive species, with *B. cereus* consistently showing higher killing efficiency. This bactericidal effect is not mediated by chemical composition but rather follows a purely physical "adhere–deform–rupture" mechanism. This mechanism has revolutionized the design paradigm of antibiofilm materials, enabling a shift from passive exclusion to an active "capture–and–kill" dual-function strategy.

**IMPORTANCE** The colonization and spread of bacteria pose significant biosafety threats to several key industries, including healthcare, food, pharmaceuticals, and biotechnology. To mitigate these risks, the current industry commonly employs intervention measures such as the addition of antibiotics, treatment with chemical disinfectants, and application of antibacterial chemical coatings. However, these chemical sterilization methods may potentially have adverse effects on human health. In contrast, the cicada wing surface, with its natural micro- and nanostructures, exhibits physical antibacterial properties that achieve efficient sterilization while avoiding the health risks associated with chemical agents, thus offering a new approach to safe antibacterial strategies.

**KEYWORDS** cicada wings, nanostructure, gram-positive bacteria, bactericidal efficiency, mechano-biocidal mechanism

The growing use of antibiotics has accelerated the development of bacterial resistance and contributed to more than 1.3 million deaths in 2019 (1–3). The discovery of new antibiotics remains slow (4), while the persistence of microbial and viral pathogens on various surfaces facilitates rapid disease transmission, which poses a serious threat to public health. In this context, natural antimicrobial surfaces that can serve as alternatives to chemical agents have gained significant interest. Among them, cicada wings have drawn attention due to their unique physical bactericidal properties. The upper and lower surfaces of cicada wings are covered with highly ordered arrays of sharp nanostructures (nanopillars, nanocones), which mechanically disrupt bacterial

Address correspondence to Liyan Wu, wly78528@syau.edu.cn.

The authors declare no conflict of interest.

integrity and kill the cells (5–10). When engineering antibacterial surfaces, focusing solely on cell-repelling forces may not constitute an optimal defense strategy; instead, a more effective approach could involve designing surfaces that actively capture and eliminate cells. Such antibacterial surfaces hold potential for application in air purification systems, where they can efficiently intercept and inactivate airborne bacteria, thereby mitigating cross-infections caused by aerosol-mediated bacterial transmission. Additionally, they may be utilized in water treatment processes to enhance water quality.

Previous studies have classified the bactericidal mechanism of nanostructures into three main types: (i) the rigidity of nanopillars causes bacteria to sink due to adhesion forces between the cell membrane and the nanopillar sidewalls, which results in direct penetration by the nanopillar tips (5, 11); (ii) as bacteria descend along the sidewalls of nanopillars, cells trapped between pillars undergo membrane deformation and rupture due to stretching rather than tip penetration (12–14); and (iii) bacterial movement across the nanopillars causes separation between the inner membrane and the lipid bilayer (gram-negative bacteria), which leads to shear deformation and membrane failure (15). Notably, current mechanistic studies still exhibit evident limitations: they focus exclusively on the geometric parameters of the nanopillars (diameter, height, spacing, and aspect ratio) and on superficial physical properties (stiffness and roughness) while overlooking other potentially critical factors (16–23).

Bacterial cellular characteristics, such as bacterial cells' mechanical stiffness and shape, represent some of the key factors impacting the bactericidal effects of nanostructures. Previous studies have explored the bactericidal effects of natural and biomimetic nanostructure materials on both gram-negative and gram-positive bacteria. Due to differences in cell rigidity, it is generally agreed that these nanostructures have more pronounced effects on gram-negative bacteria. However, the results for gram-positive bacteria remain inconsistent. Based on existing research data, a systematic comparison was conducted on the differences in bactericidal efficiency among natural biological nanostructures, silicon-based nanostructures, and gold (Au)-based nanostructures against *Staphylococcus aureus*. Hasan et al. (24) investigated the bactericidal impact of nanopatterns on the surface of cicada wings against both bacterial types. Their findings revealed that after a 30-min incubation, the surface morphology of cicada wings had a pronounced effect on gram-negative bacteria (*Branhamella catarrhalis*, *Escherichia coli*, *Pseudomonas aeruginosa*, and *Pseudomonas fluorescens*) but exhibited only minimal effects on gram-positive strains (*Bacillus subtilis*, *Planococcus maritimus*, and *S. aureus*). Shahali et al. (17) reported that the natural nanopillars significantly affected the bactericidal effect on *S. aureus* after 18 h, and *Aleeta curvicosta*'s wings showed the highest bactericidal activity, with the number of colonies dropping to $1.43 \times 10^6$ CFU/mL from an initial $2.52 \times 10^6$ CFU/mL. Bhadra et al. (16) reported that they fabricated nanopillar structures on black silicon surfaces using reactive ion etching and assessed their bactericidal efficiency against *S. aureus*. Remarkably, the nanopillar-modified silicon wafer exhibited a high bactericidal efficiency of 92% against *S. aureus* after 18 h of incubation. Nguyen et al. (18) also constructed the nanopillar structures on the surface of a silicon wafer, but the bactericidal efficiency against *S. aureus* only reached 8%. Remarkably, Wu et al. (25) fabricated Au nanopillars on surfaces by electrodepositing Au in nanoporous templates. Antibacterial assays against *S. aureus* showed that all surfaces exhibited a killing efficiency of over 99% during the 2-h interaction.

Gram-positive bacteria are showing increasingly serious drug resistance in clinical settings, and their pathogenicity poses major challenges to treatment. In this study, two representative gram-positive pathogens, *Bacillus cereus* and *S. aureus,* were selected as model organisms. These strains represent the two fundamental morphological types of gram-positive bacteria (rod-shaped and coccus-shaped, flagellated and non-flagellated) and are clinically relevant: *S. aureus* is a major cause of hospital-acquired infections (26–28), while *B. cereus* is a key pathogen in foodborne illnesses (29–32). This study investigates the cell-surface interactions between two gram-positive bacterial species and nanopillar arrays. Confocal laser scanning microscopy (CLSM) was used to quantify

live/dead cell counts following their adhesion to nanopillars, enabling the evaluation of mechanical bactericidal activity. Meanwhile, scanning electron microscopy (SEM) and transmission electron microscopy (TEM) were employed to characterize the cell-surface interface, allowing direct visualization of the nanopillar–bacteria biointerface.

## RESULTS

### Surface characterization of the cicada wings

A cicada wing consists of criss-crossing veins and thin wing membranes. Nanotopographs of the surface of the cicada wing were analyzed using atomic force microscopy (AFM) and SEM. The results indicate that hexagonal nanopillar structures were regularly distributed on the wing membrane and wing veins of the cicada wings (Fig. 1). In order to determine any correlation between the scale of the topography and antibacterial activity, a thorough analysis of the wing dimensions was required. The topographical analysis of the imaged surface structure revealed that the pitches (center-to-center distances) were within the range of 173 ± 7 nm; the heights were 367 ± 43 nm; the apex diameters were 120 ± 23 nm; the base diameters were 156 ± 29 nm; and the aspect ratio was approximately 2.35. In addition, the structure density was calculated as 41 μm$^{-2}$. The average surface roughness (Ra) and static water contact angle (WCA) were determined to be 35.4 nm and 120.8°, respectively.

### Quantitative analysis of bactericidal efficiency

After staining with SYTO9/propidium iodide (PI), the viability of *S. aureus* and *B. cereus* on the surface of cicada wings was evaluated, as shown in Fig. 2. SYTO9 is a membrane-permeable nucleic acid dye that can penetrate both intact and damaged cell membranes, bind to nucleic acids (DNA/RNA), and emit green fluorescence in both live and dead cells. In contrast, PI is membrane-impermeable and can only enter cells with

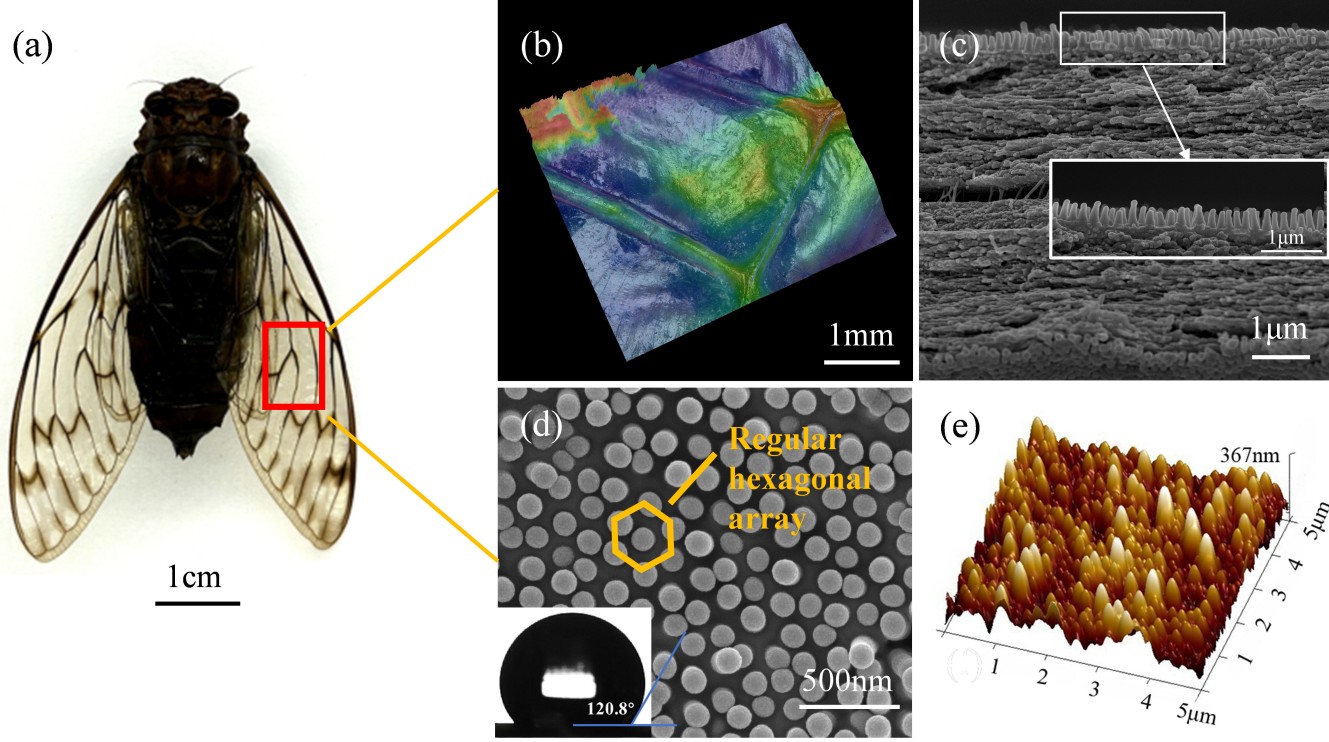

**FIG 1** Surface characterization of cicada wings. (a) Photo of *Pomponia linearis*. (b) Wing veins and membranous structure of the cicada wing. (c) Both dorsal and ventral surfaces feature nanopillar arrays. (d) SEM image showing a regular hexagonal nanopillar array on the wing surface; the lower left inset shows the water contact angle. (e) AFM image with the height cross-section profile.

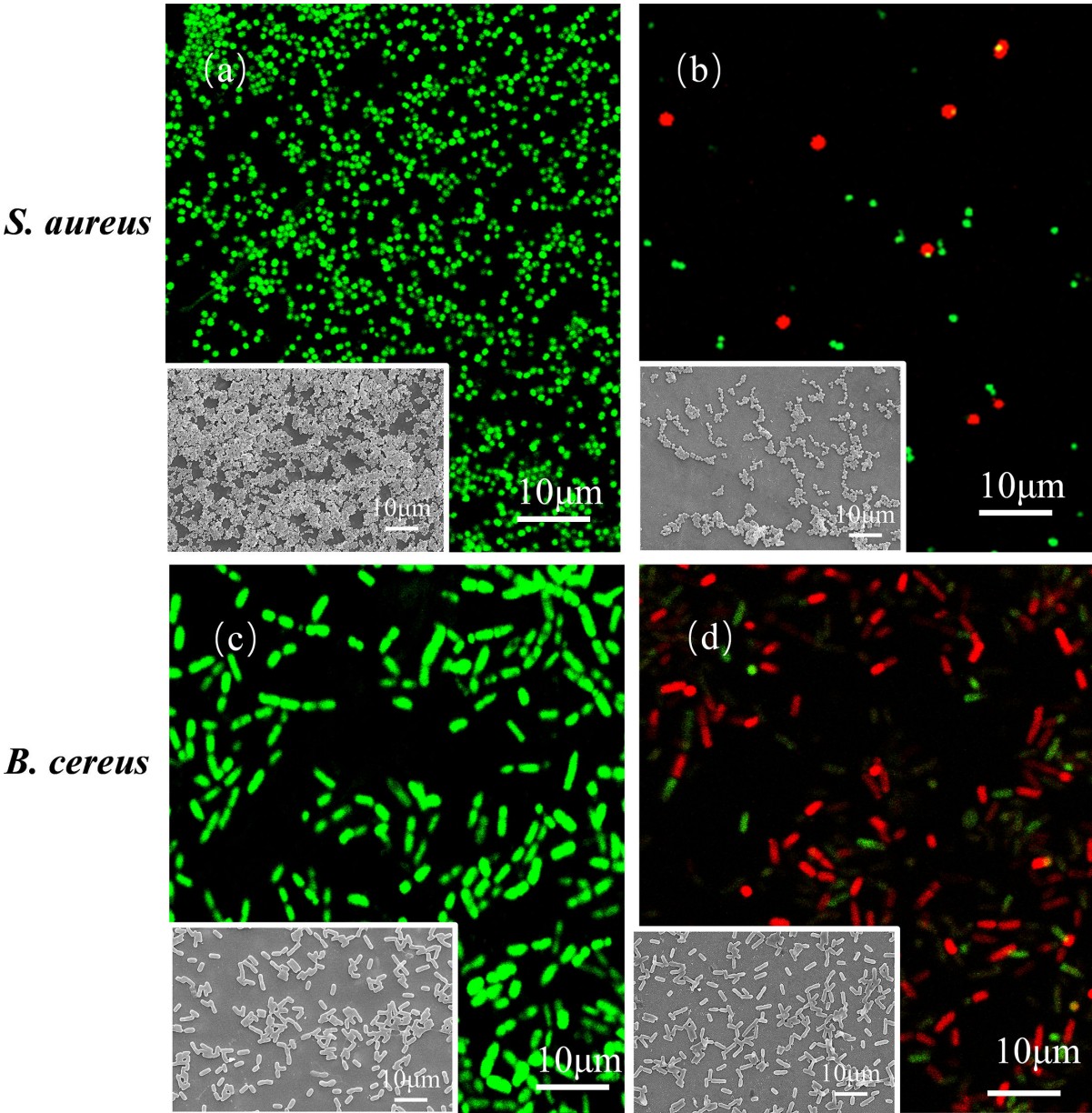

**FIG 2** Comparison of fluorescence micrographs of *S. aureus* (on blank glass control [a] and cicada wings [b]) and *B. cereus* (on blank glass control [c] and cicada wings [d]) after 24 h incubation.

damaged membranes, where it preferentially binds to double-stranded DNA and emits red fluorescence. When SYTO9 and PI are used together, PI competitively displaces the SYTO9 molecules already bound to nucleic acids. Theoretically, PI has a higher affinity for nucleic acid than SYTO9. This means that in dead cells, PI can gradually replace SYTO9, such that nucleic acid is mainly occupied by PI, which leads to red fluorescence in dead cells (33).

In Fig. S1, in addition to fluorescent green and fluorescent red signals, the presence of fluorescent yellow can be observed. This occurrence can be attributed to a partial rather than complete rupture of the bacterial cell membranes, which facilitates the entry of both PI and SYTO9, resulting in cells that are not completely stained red, indicating they are in an injured rather than fully dead state. Consequently, bacteria attached to cicada wing surfaces can be categorized into three categories: cells with intact cell membranes,

cells with partially ruptured membranes, and cells with fully ruptured membranes. This study adopted cell membrane integrity as the criterion for determining cell viability. This occurs because, once cells become permanently adhered to the cicada wing surface, the nanopillar structures exert sustained mechanical stress, which leads to complete rupture of the cell membrane and eventual cell death.

Quantitative assessment of membrane damage in two gram-positive bacterial strains using Cellpose and ImageJ revealed a significant difference in bactericidal efficiency between *B. cereus* and *S. aureus*. In the glass blank control group, there was almost no bacterial death. After 1 h of co-culture between cicada wings and bacteria, the bactericidal efficiency of the nanopillar structures against the two bacterial species was as follows: *B. cereus*: 52.7% ± 3.5% and *S. aureus*: 21.5% ± 1.9%. After 3 h of co-culture, the bactericidal efficiency against *B. cereus* (68.4% ± 0.7%) was higher than that against *S. aureus* (35.9% ± 4.2%). After 6 h of co-culture, *B. cereus* still maintained higher sensitivity (81.8% ± 1.2% vs 49.8% ± 1.3%). After 24 h of co-culture, the bactericidal efficiency of both bacteria decreased, but *B. cereus* still maintained higher sensitivity (68.7% ± 7.4% vs 23.5% ± 1.5%). The nanopillar surface exhibits potent bactericidal activity against both gram-positive species, with *B. cereus* consistently showing higher killing efficiency.

## Chemical characterization and agar-diffusion assay

Fourier transform infrared (FTIR) was used to characterize the chemical composition of the cicada wing epicuticle (34), with the spectrum shown in Fig. 3a. The peaks at 1,652 cm$^{-1}$ (C=O stretching, amide I), 1,540 cm$^{-1}$ (N–H bending, amide II), and 1,250 cm$^{-1}$ (C–N stretching combined with N–H bending, amide III) are all characteristic of chitin. The strong absorption at 2,921 cm$^{-1}$ (C–H stretching of methyl) indicates the presence of lipids, while the broad peak around 3,300 cm$^{-1}$ (O–H and N–H stretching) suggests proteinaceous components. Notably, no distinct band is observed at 1,590 cm$^{-1}$, confirming that the wing epicuticle lacks chitosan; this wavelength typically corresponds to the N–H (–NH$_2$) bending vibration of chitosan amino groups. Chitin, due to its lack of free amino groups, electroneutrality, and water insolubility, fails to effectively interact with microbial membranes, hence exhibiting no significant antibacterial activity (35, 36). In contrast, chitosan, a deacetylated product of chitin, demonstrates excellent bactericidal activity (37–39). When loaded onto nanostructured surfaces, it can achieve a synergistic effect of chemical and physical sterilization, offering new insights for the development of novel antibacterial materials.

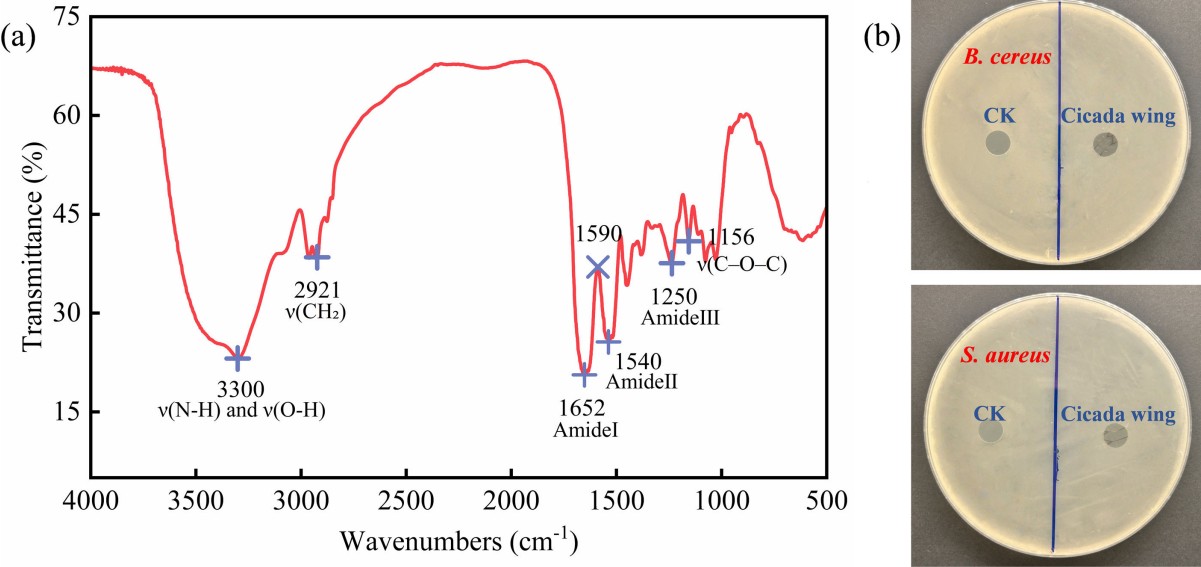

**FIG 3** Chemical analysis and antibacterial activity. (a) FTIR absorption spectrum of the cicada wing surface. (b) Agar-diffusion test results.

Figure 3b shows the results of the agar-diffusion assay. As anticipated, no inhibition zones were observed around either the glass blank control or the cicada wing, further confirming that the components of the cicada wing cuticle lack bactericidal activity. Therefore, the bactericidal effect of cicada wings primarily originates from their regularly arranged nanostructures.

## Mechano-bactericidal mechanism

Figure 4 shows original morphologies of *S. aureus* and *B. cereus*, along with SEM top views and 52° tilted views of the two bacterial cells adhering separately to cicada wing surfaces; the images visually demonstrate the dynamic process of bacteria from morphological change to cellular rupture. During the entire bactericidal process, *S. aureus* and *B. cereus* exhibit similar mechanical-deformation response phenomena. Figure 4Ia shows the intact morphology of *S. aureus*, which presents a typical perfect spherical shape with a diameter of approximately 0.68 µm. In Fig. 4Ib, orange arrows indicate deformation of cells after being interacting with nanopillars. A single *S. aureus* cell is visibly subjected to anisotropic stretching by the nanopillar array, losing its native morphology and initiating collapse downward from the pillar tips, consequently appearing significantly flatter than adjacent cells. As the nanopillars continue to exert tensile stress, the extent of bacterial deformation becomes increasingly pronounced. As depicted in Fig. 4Ic, the cell completely loses its equiaxed spherical morphology, with dimensions expanding anisotropically along specific axes. The maximum expanded dimension reaches approximately 1.12 µm, representing a 60% increase compared to the original diameter (0.68 µm). At this stage, the deformation surpasses the elastic limit of the bacterial cell, inducing plastic rupture. Ultimately, the bacterial cell collapses entirely and becomes embedded within the nanopillars' interstices, accompanied by cytoplasmic

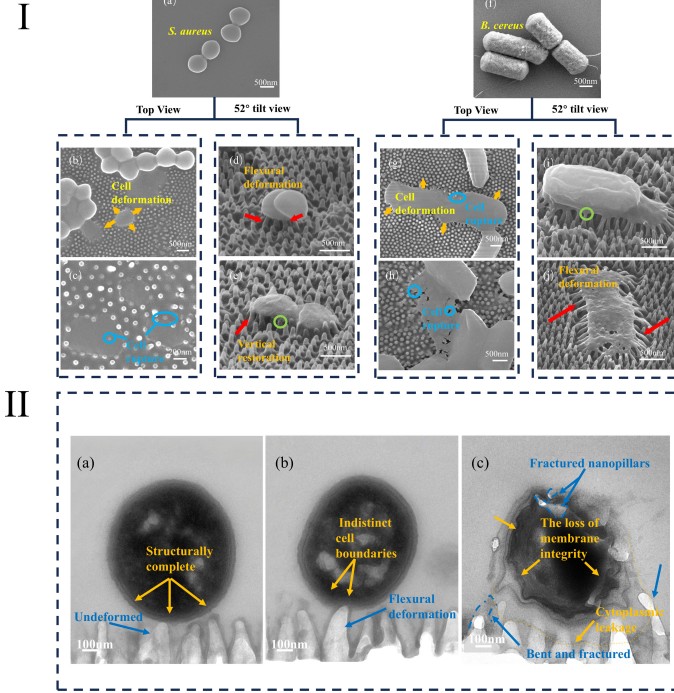

**FIG 4** Deformation of bacterial cells and nanopillars. (I) SEM imaging of bacterial cells and nanopillar. (a) Original cell morphology of *S. aureus*. (b and c) Top-view images of *S. aureus* and nanopillar deformation. (d and e) 52° tilt-view images of *S. aureus* and nanopillar deformation. (f) Original cell morphology of *B. cereus*. (g and h) Top-view images of *B. cereus* and nanopillar deformation. (i and j) 52° tilt-view images of *B. cereus* and nanopillar deformation. (II) TEM imaging of bacterial cells and nanopillar. (a–c) Process of nanopillar–bacterium contact, deformation, and rupture.

leakage due to membrane permeabilization. The same process can also be observed in *B. cereus* cells attached to the cicada wing surfaces (Fig. 4If through Ih). Under the mechanical influence of the nanopillar structures, *B. cereus* cells undergo anisotropic stretching and deformation. The deformation in the width direction is significantly more pronounced than that in the length direction, with the width dimension increasing by 65% relative to the original diameter (0.8 µm). *B. cereus* cells also exhibit a progressive structural collapse initiated at the apices of the nanopillars, ultimately resulting in cytoplasmic leakage.

Rupture was observed in both the peripheral and non-peripheral regions of the bacterial cell in both bacterial species. Notably, the cell-disruption processes differed between two the bacterial species: *B. cereus* exhibited more extensive structural damage, with significantly larger damaged-area ratios compared to *S. aureus*. In contrast, loss of cell integrity in *S. aureus* was predominantly confined to the peripheral region. These findings are fully consistent with previous reports (22, 40) and are attributed to the difference in cell rigidity between these two gram-positive bacteria.

In the top-view image, the nanopillars exhibit pronounced axial elongation, with the extension reaching 1.5–2.0 times their original length and accompanied by evident cross-sectional shrinkage, manifesting distinct tensile deformation characteristics.

The deformation characteristics of bacterial cells and the deformation behavior of nanopillars in the tilted views and top views were comparatively analyzed. In agreement with the top-view images, the nanopillars also primarily exhibit tensile deformation. Additionally, the tilted view reveals that local constraint from the overlying cell membrane introduces a radial bending component: pillar tips deflect toward the cell center, while their bases remain fixed, forming a geometry consistent with the elastic deflection of a cantilever beam under transverse end loading. At the cell periphery, all nanopillars are bent inward, from the cell edge toward the center in Fig. 4Id and Ij. As highlighted by the red arrows in Fig. 4Ie, the stress concentration releases the constraint on the pillars, allowing bent pillars to relax to their initial position. This deflection–recovery sequence reveals that intense pulling forces operate across the bacterium–interface system: the pillars act as miniature cantilevers, storing elastic strain energy that is subsequently transferred to the cell membrane, accelerating its destabilization and lysis.

Tilted-view images reveal that the plasma membrane is visibly "pulled down" at each nanopillar contact point, giving rise to a locally protruding, pouch-like invagination. These downward bulges exhibit an average area of 0.010–0.015 µm² ($n = 30$) and are highlighted by the green circle in Fig. 4Ie and i. In stark contrast, the membrane spanning the gaps between adjacent nanopillars remains comparatively flat, showing no appreciable change in curvature. These tensile forces create uneven membrane topography around each nanopillar, giving rise to pronounced local curvature effects. The uneven top surface of the cell suggests loss of turgor pressure related to cell rupture.

SEM images primarily visualize the deflected profiles of cell-edge regions and adjacent nanopillars; however, the interaction dynamics between the cell base and underlying pillars remain obscure. To address this, we employed TEM to capture *in situ* deformation across the entire bacterium–substrate interface. At the initial contact stage between the bacterium and the substrate, Fig. 4IIa shows an intact cellular architecture: the cell is sharply outlined and smooth; the cell retains a perfectly round profile, indicating a healthy state; and the nanopillars remain upright and undeformed. Upon transition to irreversible adhesion (Fig. 4IIb), the pillars beneath the bacterium bend, and the boundaries between the cell wall and the membrane become indistinct. Figure 4IIc clearly reveals that the bacterium has lost its structural integrity: the cell is ruptured, and in the still-intact regions, it appears wrinkled. Cytoplasmic leakage is visible around the nanopillars beneath the cell, and at the site of most severe disruption, fragments of broken pillars can be seen. A pronounced tensile force exerted across the nanopillar–cell interface has thus compromised cellular integrity.

Figure S2 captures a snapshot of cell division on the nanopillar surface for both bacterial species. The process commences with the assembly of a dynamic Z-ring by

FtsZ protein at the prospective division site. Septal peptidoglycan synthases associated with the ring drive its progressive constriction, with the resulting mechanical tension transmitted across transmembrane proteins to the peptidoglycan layer, the major load-bearing component of the gram-positive cell wall (41, 42). This radial, inward pull induces viscoelastic deformation of the wall, generating a characteristic invagination. As the local radius of curvature decreases, the surface tension of the peptidoglycan layer increases, creating a stretched, tensed state that ultimately triggers cell constriction.

## DISCUSSION

### Differences in bactericidal efficiency

In this study, the natural nanopillar array on the wing surface of the cicada *Pomponia linearis* exhibited distinct physical bactericidal activity against both *S. aureus* and *B. cereus*. During the initial contact phase (1–6 h), the bactericidal efficacy increased significantly with time; however, after extending the co-incubation to 24 h, the efficacy declined. This phenomenon can be attributed to (i) a progressive increase in bacterial adhesion, with cells accumulating into multilayered stacks that shield upper-layer bacteria from direct contact with the nanopillars, thereby diminishing the physical bactericidal efficacy; and (ii) the secretion of extracellular polymeric substances during prolonged attachment, which gradually forms a dense biofilm over the nanopillar surface. The biofilm not only covers the nanopillar structures but also provides protection for the bacteria, hindering the direct contact between nanopillars and bacteria and further inhibiting the bactericidal effect.

Despite the identical nanostructured substrates, we observe a significant difference in bactericidal effects for the two bacterial species. The difference in the degree of cell rupture between the two bacteria may be closely related to the inherent characteristics of the bacteria themselves (e.g., their morphology and cell rigidity). The projected area of *B. cereus* on the nanostructured substrate is 2.21–7.71 times that of *S. aureus*. As a result, the cells are anchored by more nanopillars and experience higher mechanical tension, which increases the likelihood of cell damage. The cell envelope of gram-positive bacteria is a rigid external structure, serving as the first barrier between the bacterium and its environment. It maintains cellular shape and withstands the osmotic pressure generated by cell turgor. An increase in cell wall thickness is typically accompanied by enhanced rigidity. Previous studies comparing the bactericidal effects of cicada wings on gram-negative and gram-positive bacteria have shown that gram-positive cells are generally more rigid and resistant to mechanical lysis, owing to a peptidoglycan layer that is four to five times thicker than that of gram-negative bacteria. This greater cell wall thickness requires a greater deformational stress to disrupt the cell wall, deform the inner membrane, and cause cell death (40, 43). Currently, research data on the cell wall thickness of *B. cereus* is extremely limited, with virtually no direct measurement results available for reference. However, several studies indicate that ultrastructural analysis of live *S. aureus* cells indicates their cell wall thickness of approximately 20 nm (44). In contrast, *B. subtilis* exhibits a cell wall thickness of ~33 nm (45). Notably, under hydrated conditions, the cell wall thickness of the two species shows minimal distinction: *B. subtilis* (36 ± 5.3 nm) vs *S. aureus* (34 ± 10 nm) (46). Given the high similarity in cell wall structure between *B. subtilis* and *B. cereus*, it is inferred that the cell wall thickness of *B. cereus* is comparable to that of *S. aureus*. In addition to cell wall thickness, Young's modulus serves as a key indicator of stiffness; *B. cereus* exhibits a value roughly 10-fold that of *S. aureus* (47, 48). Nevertheless, the former is killed more efficiently, a consequence of its larger, rod-like geometry, which generates greater local deformation imposed by the nanopillars and thereby exceeds the critical rupture threshold.

### Mechano-biocidal mechanisms

The mechano-bactericidal action of nanostructures is independent of surface chemistry: the cicada wing surface composition is primarily chitin, a material that has previously

been shown to possess virtually no intrinsic antimicrobial activity. In prior studies, 6–10 nm gold layers were sputtered onto cicada, dragonfly, or black-silicon surfaces. This treatment modified surface chemistry while leaving the nanopillar architecture essentially intact, and bactericidal efficacy remained unaltered, confirming that the killing mechanism is purely physical rather than chemical (5). Upon contact with the nanopatterned substrate, both bacterial species—driven by thermodynamics and adhesion mechanisms—reorient themselves to maximize contact area with the surface, thereby minimizing system free energy and consolidating adhesive stability (49). The bacterial cell envelope can be regarded as an elastic membrane. Upon adhesion to the nanopillar surface, it contracts to maintain overall structural stability, thereby generating tensile forces between the cell envelope and the extracellular nanostructures. The cell is stretched, while the nanopillars bend and elongate like miniature cantilevers. During this mutual interaction, both components gradually approach their respective mechanical limits, eventually leading to bacterial cell failure. Although nanopillar fracture is occasionally observed under experimental conditions, its probability remains far lower than that of cell rupture. Cell division further intensifies this process by driving additional contraction of the bacterium on the substrate, exacerbating the tensile interplay and accelerating bacterial cell destabilization. Lohmann et al. (50) have reported similar nanopillar deflection phenomena, attributing them to bacterial motility. However, if bacterial movement were the sole cause, nanopillars would not exhibit consistent bending deformations toward the cell center. In the case of mere overall bacterial displacement, nanopillars should show randomly oriented and time-varying deflections rather than sustained, radially symmetric bending toward

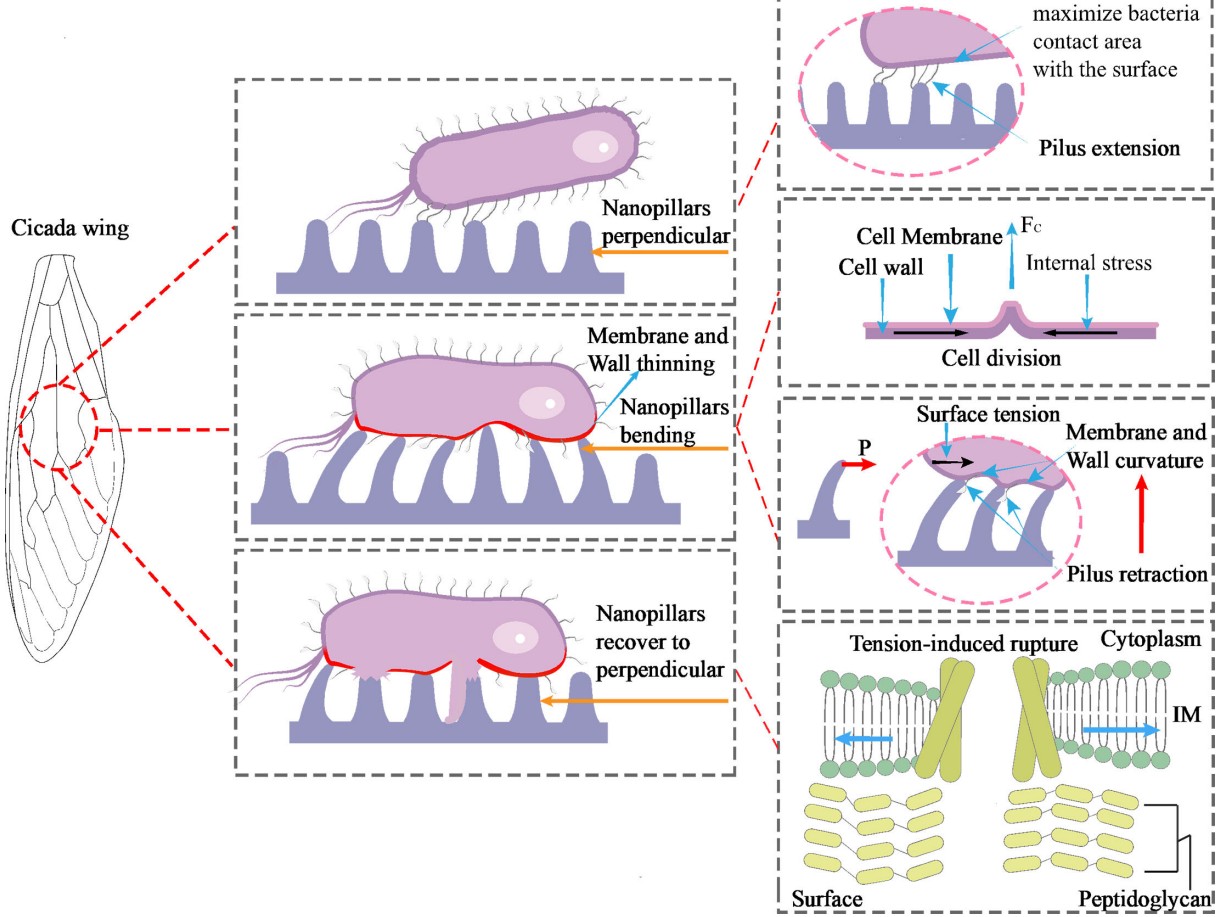

**FIG 5** Schematic diagram of the mechano-bactericidal mechanism.

the cell center. Moreover, bacterial surface appendages—flagella and pili—constitute a critical mechanical anchoring system during early colonization: flagella first extend and anchor to achieve initial adhesion to the substrate. Upon attachment, pili retract within the directional tensile field provided by the nanopillar array, markedly increasing local membrane curvature. As cell envelope tension and curvature change, transmembrane helices of mechano-sensitive channel proteins are pulled apart, ultimately leading to cell body rupture to achieve the interface-induced "assisted rupture" effect (Fig. 5).

## Conclusions

For the first time, we systematically examined how the three-dimensional surface topography of *P. linearis* cicada wings influences the adhesion and bactericidal activity against gram-positive bacteria. Experimental results show that both *B. cereus* and *S. aureus* can be killed on the cicada wing surface, and the number of dead *B. cereus* cells is statistically significantly higher than that of *S. aureus* cells. This difference is mainly attributed to the heterogeneity in the thickness of the peptidoglycan layer, cell wall elastic modulus, and overall cell morphology between the two bacteria. Chemical characterization, combined with antibacterial assays, confirmed that the cicada wing surface is composed almost entirely of chitin and that its bactericidal action does not rely on any chemical mechanism; killing is effected solely by physical interaction. In both SEM and TEM images, elastic deformation can be observed after bacteria come into contact with nanopillars. As the cell contracts, nanopillars bend and axially elongate under lateral tensile forces, while the bacterial envelope undergoes local stretching, leading to reduced membrane thickness and the formation of local high-curvature regions at the contact points with nanopillars. Once the local membrane strain exceeds its elastic limit, irreversible mechanical rupture occurs. Notably, the contraction of the cell membrane during division further promotes cell rupture. Future work should establish a multiscale mechanical model through multiphysics coupling simulations to systematically resolve the dynamic interplay between nanopillar arrays and the cell. By quantifying the contribution of each force component, such modeling should provide a mechanistic, quantitative description of how micro-/nanopillared surfaces inflict mechanical damage, offering theoretical guidance for the precision design of next-generation antibacterial materials.

## MATERIALS AND METHODS

### Cicada wing preparation

Cicada specimens (*P. linearis*) were purchased online, with their origin traced to Wuhan, China (Fig. 1a). Wings from undamaged adult female specimens were cut into uniform 5 mm disks using a biopsy punch (Kai Medical Biopsy Punch; Kai Industries Co. Ltd., Seki, Japan). The wing samples were ultrasonically cleaned in deionized water for 10 min, followed by surface sterilization in 75% ethanol for 8–10 min to remove organic residues and surface-associated bacteria. The samples were then air-dried at room temperature inside a laminar flow cabinet and then stored at 4°C until use. All experiments were performed on the same region of the forewing to ensure consistency.

### SEM

In order to accurately obtain the topographical feature data (including nanopillar spacing, density, and diameter), measurements were conducted using a SEM (Regulus 8100; Hitachi, Japan) with an acceleration voltage of 5 kV and magnifications of ×10,000, ×30,000, and ×50,000.

Images of bacterial morphology were taken using a Hitachi-Regulus 8100 SEM at 3 kV under ×5,000, ×10,000, and ×20,000 magnifications.

**TABLE 1** Bacterial traits of the strains used in this study

| Bacterial strain | Origin | Cell shape | Size (µm) | Motility | Gram type | Pathogenicity |
|---|---|---|---|---|---|---|
| *B. cereus* | Isolated from cicada wings | Rod | Length: 1.0–3.5 Width: 0.8 | Peritrichously flagellated | Positive | Pathogenic |
| *S. aureus* | ATCC 25923 (Xinagf Bio, Shanghai) | Coccus | Diameter: 0.68 | Non-flagellated | Positive | Pathogenic |

## AFM

The topography height images of cicada wings were obtained using an AFM (Dimension Edge; Bruker, Germany) in tapping mode at 2 Hz mounted on air-dried mica substrates. Scans were conducted using phosphorus-doped silicon probes (RTESP300, Bruker) with a resonant frequency of 300 kHz, tips with a radius of curvature of 15 nm, and a scan range of 5 µm × 5 µm. Images were analyzed using the analysis software NanoScope Analysis.

## Water contact angle

The static WCA was measured using a contact angle meter (OCA200; Dataphysics, Germany). Droplets of 3 µL of deionized water were used for the WCA analysis, at a dispensing rate of 1 µL/s. For each sample, five replicate measurements were conducted, and the final data were the mean values of these measurements.

## Bacterial culture

The selected model bacterial strains for this study were *B. cereus* and *S. aureus*, as shown in Table 1. Bacteria from glycerol stocks were cultivated on Luria–Bertani plates (LB). A single colony was transferred to 50 mL of LB broth and incubated overnight at 37°C with shaking at 180 rpm. An aliquot of 1 mL from the overnight culture was added to 100 mL of fresh LB medium and incubated for 8–10 h to serve as the experimental bacterial suspension.

## CLSM

The bacterial suspension was harvested via centrifugation at 4,500 rpm for 10 min, and the cell pellet was resuspended in 0.85% NaCl to an $OD_{600}$ of 0.7 (8). The cicada wings were co-cultured with 5 mL of suspension per well in six-well plates for 1, 3, 6, and 24 h at 37°C. As a control group, sterile glass slides were added to equal volumes of bacterial suspension, with all other conditions kept identical to the experimental group. The wings were removed and washed twice with 0.85% NaCl to remove loosely bacteria. A field-emission SEM (Apreo 2C; Thermo Fisher Scientific, Waltham, MA, USA) at 20 kV was used to obtain high-resolution electron micrographs of cicada wings with adhering bacteria at magnifications of ×5,000 and ×10,000.

The SYTO9/PI Live/Dead Bacterial Double Stain Kit (MX4234; MKbio, Shanghai, China) was used to assess cell membrane integrity. A staining solution was prepared by mixing 1 mL of 0.85% NaCl with 1.5 µL SYTO9 (excitation/emission: 480 nm/500 nm) and 1.5 µL PI (excitation/emission: 535 nm/617 nm). Cicada wing samples were immersed in the staining solution for 10 min in the dark to avoid photobleaching. Live and dead bacteria were visualized using an inverted confocal laser scanning microscope (Leica TCS SP8; Leica, Wetzlar, Germany) equipped with a ×63/NA 1.45 oil immersion objective. Images were acquired from three to four randomly selected fields of view to ensure statistical relevance. In addition, all experiments were independently repeated at least twice to confirm reproducibility. Bacterial viability was quantified using Cellpose (51, 52) and ImageJ software. The dead cell ratio was calculated as follows (53):

$$\text{Bactericidal efficiency (\%)} = \frac{\text{Number of membrane disrupted cells}}{\text{Number of adhered cells}} \times 100\% . \tag{1}$$

## TEM

Cicada wings co-cultured with bacteria for 24 h were used as samples. Samples were prepared by washing with 0.01 M PBS and post-fixed 0.1 M cacodylate buffer twice for 10 min and post-fixed with 1% osmium tetroxide in a cacodylate buffer for 1 h. The samples were washed with UHQ water for 10 min three times and immersed in 1% uranyl acetate in water for 1 h. These samples were dehydrated with 30%, 50%, 60%, 70% 90%, and 100% ethanol for 10 min each. Samples were embedded in fresh resin and polymerized at 70°C for 48 h before slicing and imaging the interface under TEM (HT7800, Hitachi).

## FTIR microspectroscopy

FTIR is useful for investigating the chemical composition of defined areas of materials. A surface chemical characterization of the cicada wing surfaces was performed using the FTIR microscope equipped (Nicolet Summit LITE, Thermo Fisher Scientific, Waltham, MA, USA) with a single-bounce ATR accessory (Nicolet iS50, Thermo Fisher Scientific, Waltham, MA, USA). The spectra were collected at 0.125 cm$^{-1}$ spectral resolution with 16 scans co-added for each measurement position, with the time required for map collection being approximately 1 s per pixel. Data were collected over a wavenumber range of 4,000–1,000 cm$^{-1}$.

## Agar-diffusion assay

A standard bacterial suspension ($10^8$ CFU/mL) was evenly spread on an LB agar plate using a sterile swab to create a bacterial lawn. The cicada wing sample was then aseptically placed on the agar using sterile forceps, with a sterilized glass disk of equal diameter serving as a blank control. The plates were incubated at 37°C for 24 h. Subsequently, the zones of inhibition were observed, and their diameters were measured with a caliper. All experiments were performed in triplicate.

### ACKNOWLEDGMENTS

J.Q.: conceptualization, methodology, investigation, formal analysis, and writing–original draft.; S.G.: formal analysis, investigation, and data curation; L.C.: methodology and visualization; F.C.: data curation; H.W.: project administration and writing–review & editing; L.W.: conceptualization, supervision, and writing–review & editing.

The authors declare that they have no known competing financial interests or personal relationships that could have appeared to influence the work reported in this paper.

This work was financially supported by the Innovation Team Project of Department of Education of Liaoning Province, PRC (Grant No. LJ222510157002), and the Key Research and Development Project of Liaoning Province (Grant No. 2024JH2/102500004).

### AUTHOR AFFILIATION

[1]Shenyang Agricultural University, Shenyang, China

### AUTHOR ORCIDs

Jianwei Qu http://orcid.org/0009-0004-4442-3328
Liyan Wu http://orcid.org/0000-0003-1491-5204

### AUTHOR CONTRIBUTIONS

Jianwei Qu, Conceptualization, Formal analysis, Investigation, Methodology, Writing – original draft | Shiya Gu, Data curation, Formal analysis, Investigation | Lei Chen, Methodology, Visualization | Fangming Cui, Data curation | Huan Wang, Project administration, Writing – review and editing | Liyan Wu, Conceptualization, Supervision, Writing – review and editing

## ADDITIONAL FILES

The following material is available online.

### Supplemental Material

**Supplemental material (Spectrum02037-25-s0001.docx).** Fig. S1 and S2.

### Open Peer Review

**PEER REVIEW HISTORY (review-history.pdf).** An accounting of the reviewer comments and feedback.

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
