## [Reviewer comments · Microbiology Spectrum]

Microbiology Spectrum

Mechano-Bactericidal Activity of Cicada Wing Nanostructures Against Gram-Positive Bacteria

Jianwei Qu, Shiya Gu, Lei Chen, Fangming Cui, Huan Wang, and Liyan Wu

Corresponding Author(s): Liyan Wu, Shenyang Agricultural University

Review Timeline:

Submission Date:	July 4, 2025
Editorial Decision:	August 17, 2025
Revision Received:	October 15, 2025
Accepted:	November 13, 2025

Editor: Valeria Allizond

Reviewer(s): The reviewers have opted to remain anonymous.

Transaction Report:

DOI: <https://doi.org/10.1128/spectrum.02037-25>

Re: Spectrum02037-25 (Mechano-Bactericidal Activity of Cicada Wing Nanostructures Against Gram-Positive Bacteria)

Dear Ms. Jianwei Qu:

Thank you for the privilege of reviewing your work. Below you will find my comments, instructions from the Spectrum editorial office, and the reviewer comments.

As the Editor, I strongly recommend that the authors revise the manuscript according to Reviewer 1's comments, ensuring greater completeness and detail, before resubmitting it to the journal. Of course, the indications from Reviewer 2 should also be addressed.

Revision Guidelines

Sincerely,
Valeria Allizond
Editor
Microbiology Spectrum

Reviewer #1 (Comments for the Author):

The manuscript of Qu et al. is dedicated to investigations of antibacterial and antibiofilm properties of cicada wings surface and

a potential mechanism of this action. Also, authors suggest those surface structures are prospective for future industrial application.

Regretfully, in the form in which the manuscript exists now, it should be fundamentally rewritten. Here there are reasons why:

1) Insect wings normally contain chitin as a major cuticle component, and the wing surface includes nanofibers and nanopores (for instance <https://www.sciencedirect.com/science/article/pii/S0141813015000185>)

2) It is well-known that chitin and its derivatives have antibacterial and antibiofilm activity (for instance, <https://www.sciencedirect.com/science/article/pii/S014181301836015X?via%3Dihub>; <https://journals.plos.org/plosone/article?id=10.1371/journal.pone.0189537>; <https://www.sciencedirect.com/science/article/pii/S0304416505003028>).

There were also attempts to make chitosan-covered surfaces to prevent bacterial contamination (<https://www.tandfonline.com/doi/abs/10.1163/156856208784909372> or <https://pubs.acs.org/doi/abs/10.1021/acs.biomac.0c00127>).

In this way, it is strange to read a manuscript describing the effect and the mechanism of chitin/chitosan action without any mention of chitin/chitosan itself. Authors should better investigate the field they are working in.

Another major point is when providing the microscopy images, it would be better to use controls. And not only for microscopy of course. For instance, authors declare the absence of microbial growth on the wing surface at different time points, and there should be an image attached to each wing image that contains the biofilm on the control surface (glass, polystyrene, or any other).

So, it is highly recommended for authors to decide what exactly they investigate. Chitin is a rather well-studied compound as an antimicrobial agent, and maybe it is useful to go not into the study of cicada wings themselves, but maybe of some surfaces and chitin surface organization similar to cicada wings? Anyway, this work is consistent and well-done, but unfortunately it repeats the known facts and ignores them.

Reviewer #2 (Comments for the Author):

Overall in this study the authors investigated the adhesion behavior and antibacterial properties of two Gram-positive bacterial strains on the surface of cicada wings. The paper is well written and fascinating.

The authors use of SYTO9 and PI to show membrane integrity and dead cells was appropriate and shows cells with intact membranes, cells with partially ruptured membranes, and cells with fully ruptured membranes.

Figure 3 is very telling and shows strong bactericidal results

Figure 4 is good in that it shows the mechanism responsible.

Line 80 - italicize *Staph aureus*.

A little more discussion on the differences observed between Gram-positive and Gram-negative bacteria would be useful in the conclusion section.

The manuscript of Qu et al. is dedicated to investigations of antibacterial and antibiofilm properties of cicada wings surface and a potential mechanism of this action. Also, authors suggest those surface structures are prospective for future industrial application.

Regretfully, in the form in which the manuscript exists now, it should be fundamentally rewritten. Here there are reasons why:

1) Insect wings normally contain chitin as a major cuticle component, and the wing surface includes nanofibers and nanopores (for instance

<https://www.sciencedirect.com/science/article/pii/S0141813015000185>)

2) It is well-known that chitin and its derivatives have antibacterial and antibiofilm activity (for instance, <https://www.sciencedirect.com/science/article/pii/S014181301836015X?via%3Dihub>;

<https://journals.plos.org/plosone/article?id=10.1371/journal.pone.0189537>;

<https://www.sciencedirect.com/science/article/pii/S0304416505003028>).

There were also attempts to make chitosan-covered surfaces to prevent bacterial contamination

(<https://www.tandfonline.com/doi/abs/10.1163/156856208784909372> or

<https://pubs.acs.org/doi/abs/10.1021/acs.biomac.0c00127>).

In this way, it is strange to read a manuscript describing the effect and the mechanism of chitin/chitosan action without any mention of chitin/chitosan itself. Authors should better investigate the field they are working in.

Another major point is when providing the microscopy images, it would be better to use controls. And not only for microscopy of course. For instance, authors declare the absence of microbial growth on the wing surface at different time points, and there should be an image attached to each wing image that contains the biofilm on the control surface (glass, polystyrene, or any other).

So, it is highly recommended for authors to decide what exactly they investigate. Chitin is a rather well-studied compound as an antimicrobial agent, and maybe it is useful to go not into the study of cicada wings themselves, but maybe of some surfaces and chitin surface organization similar to cicada wings? Anyway, this work is consistent and well-done, but unfortunately it repeats the known facts and ignores them.

Response to the Reviewers

Manuscript# Spectrum02037-25

Title: Mechano-Bactericidal Activity of Cicada Wing Nanostructures Against Gram-Positive Bacteria

Dear Reviewers, we would like to thank you for your careful reading and constructive suggestions. We have revised the manuscript in accordance with your excellent advises. The point-to-point replies are listed as followed. Now we submit it to you for further consideration of publication. Please see more details in our revision, the changes we have made are highlighted with **red color** in the marked revised manuscript. We hope the revised manuscript can satisfy you and meet the high standard requirement for the esteemed Journal. Thank you very much for your attention and consideration.

Followings are the point-by-point responses (in Regular black font) to the comments (in *Italic blue font*) from Reviewer.

Thanks again for your kind comments.

Response to Reviewer #1:

Comment 1: *Insect wings normally contain chitin as a major cuticle component, and the wing surface includes nanofibers and nanopores.*

Response: Thank you very much for your valuable comments. Indeed, chitin serves as the primary structural component in insect wings. We have supplemented our study with FTIR experiments to determine the chemical composition of the cicada wing surface, as shown in Fig. 1. Please see Page 9, Line 162-176 in the 'Marked-Up Manuscript' file.

Fig. 1. FTIR analysis of the wing surfaces.

The characteristic peaks at 1652 cm⁻¹ (C=O stretching, amide I), 1540 cm⁻¹ (N–H bending, amide II), and 1250 cm⁻¹ (C–N stretching combined with N–H bending, amide III) confirm that the cicada wings used in this research are composed of chitin. Although insect wings share the same chemical composition—chitin—their surface microstructures vary significantly across species, exhibiting remarkable structural diversity. For example: (1) The surface of cicada wings displays a regular array of nanopillars (Fig. 2). These nanopillars possess specific height, diameter, and spacing, endowing the wings with excellent superhydrophobicity and physical antibacterial properties; (2) Butterfly wings are covered with scale-like structures (Fig. 3) that carry microscopic grooves, giving them anisotropic wettability and self-cleaning ability.

Fig. 2. Nanotopographs of the surface of the cicada wings.

Fig. 3. Nanotopographs of the surface of the butterfly wings. The image is reproduced from Microstructure and structural color in wing scales of butterfly *Thaumantis diores* (<https://doi.org/10.1007/s11434-009-0076-8>), with permission from the original publisher.

Although the chemical composition of insect wings is consistent, the regulation of surface micro-nano structures—including their morphology, dimensions, and arrangement—enables the realization of vastly different physical, chemical, and biological functions. We have incorporated the above points into the revised manuscript and thank you again for your thorough review and valuable suggestions.

Comment 2: It is well-known that chitin and its derivatives have antibacterial and antibiofilm activity.

Response: Thank you very much for your valuable comments. FTIR analysis confirmed the presence of methylene, amide, carbonyl groups, as well as OH stretching vibrations associated with chitin, waxes, and carboxylic acids. This is consistent with the findings of Shahali's study (<https://doi.org/10.1039/C8TB03295E>). Previous studies have shown that chitin cannot effectively interact with microbial membranes, hence exhibiting no significant antibacterial activity. (<https://doi.org/10.5171/2013.369217>)

Native chitin itself has almost no antibacterial activity and must be chemically modified (such as deacetylation, quaternization, carboxymethylation, etc.) or combined with nano-silver, antibiotics, etc., to confer significant antibacterial properties.

To verify the above analysis, we supplemented an agar diffusion experiment using the *S. aureus* and the *B. cereus*. Fig. 4 shows that under the culture conditions of 24 hours and 37 °C, no antibacterial zones (inhibition diameter = 0 mm) appeared around the cicada wing samples and the blank glass slides. This experiment further supports the conclusion that "The mechano-bactericidal action of nanostructures is independent of surface chemistry."

Fig. 4. Agar-diffusion test results.

Thank you for providing the literatures. We agree with the general consensus that "chitin derivatives (such as chitosan) have excellent antibacterial properties," but in this work, only native chitin was detected on the surface of cicada wings, and its chemical structure determines that it has no significant antibacterial effect. We have made a clear explanation on pages 15-16 of the revised manuscript.

Thank you again for your meticulous review and constructive suggestions, which have been of great benefit to improving the rigor of the manuscript. Please see Page 10, Line 177-183 in the 'Marked-Up Manuscript' file.

Comment 3: There were also attempts to make chitosan-covered surfaces to prevent bacterial contamination.

Response: Thank you for your insightful suggestion, which is highly enlightening and opens new pathways for our subsequent research. Chitosan coatings have been extensively demonstrated to possess broad-spectrum antibacterial properties. In future work, we plan to integrate the chemical activity of chitosan with the physical nanostructures of insect wings to develop a synergistic, multi-modal antibacterial surface. This approach is expected to enhance antibacterial performance in functional

material design.

Comment 4: Another major point is when providing the microscopy images, it would be better to use controls. And not only for microscopy of course. For instance, authors declare the absence of microbial growth on the wing surface at different time points, and there should be an image attached to each wing image that contains the biofilm on the control surface (glass, polystyrene, or any other).

Response: Thank you very much for your constructive suggestion. Following your advice, we have added glass slides of the same size as blank controls during both SEM and CLSM imaging. We have added the aforementioned control images and quantitative data to the revised manuscript, and the “Materials and Methods” section now includes a detailed description of the control setup. Thank you again for your insistence on experimental rigor—it greatly enhances the reproducibility and persuasiveness of our work. Please see Page 7, Line 134 and Supplementary Information (Fig.S1) in the 'Marked-Up Manuscript' file.

Response to Reviewer #2:

Comment 1: Line 80 - italicize *Staph aureus*.

Response: Thank you for your reminder. We have italicized "*Staphylococcus aureus*" on line 80 of the main text and checked the entire manuscript to ensure that all bacterial names conform to microbial nomenclature conventions. Owing to layout adjustments, this phrase now appears on line 74.

Comment 2: A little more discussion on the differences observed between Gram-positive and Gram-negative bacteria would be useful in the conclusion section.

Response: Thank you for your valuable suggestions. We have expanded the discussion on the differences between Gram-positive and Gram-negative bacteria in the manuscript. This study initially focused on Gram-positive bacteria because their thick and directly exposed peptidoglycan layer presents a distinctive mechanical response model for nanostructured surfaces.

Regarding the impact on Gram-negative bacteria, we fully acknowledge its importance. Gram-positive cells are generally more rigid and more resistant to mechanical lysis than Gram-negative cells due to their peptidoglycan cell wall being 4 - 5 times thicker than that of Gram-negative bacteria. This greater cell wall thickness requires greater deformational stress to disrupt the cell wall, deform the inner membrane, and cause cell death. Please see Page 16, Line 294-298 in the 'Marked-Up Manuscript' file.

Re: Spectrum02037-25R1 (Mechano-Bactericidal Activity of Cicada Wing Nanostructures Against Gram-Positive Bacteria)

Dear Prof. Liyan Wu:

Your manuscript has been accepted, and I am forwarding it to the ASM production staff for publication. Your paper will first be checked to make sure all elements meet the technical requirements. ASM staff will contact you if anything needs to be revised before copyediting and production can begin. Otherwise, you will be notified when your proofs are ready to be viewed.

Sincerely,
Valeria Allizond
Editor
Microbiology Spectrum

Reviewer #1 (Comments for the Author):

The article now is much better, actually it is fine. The only thing I would recommend is to fix on the figure 3 the "S. aureu" to "S. aureus".

Thank you!